# A Silicon-Based On-Chip 64-Channel Hybrid Wavelength- and Mode-Division (de)Multiplexer

Yuxiang Yin [1,2], Hang Yu [1,2], Donghe Tu [1,2], Xingrui Huang [1,2], Zhiguo Yu [1], Huan Guan [1,*] and Zhiyong Li [1]

1   State Key Laboratory on Integrated Optoelectronics, Institute of Semiconductors, Chinese Academy of Sciences, Beijing 100083, China
2   College of Materials Science and Opto-Electronic Technology, University of Chinese Academy of Sciences, Beijing 100083, China
*   Correspondence: yiyigh@semi.ac.cn

**Abstract:** An on-chip 64-channel hybrid (de)multiplexer for wavelength-division multiplexing (WDM) and mode-division multiplexing (MDM) is designed and demonstrated on a 220 nm SOI platform for the demands of large capacity optical interconnections. The designed hybrid (de)multiplexer includes a 4-channel mode (de)multiplexer and 16-channel wavelength-division (de)multiplexers. The mode (de)multiplexer is comprised of cascaded asymmetric directional couplers supporting coupling between fundamental TE mode and higher-order modes with low crosstalks in a wide wavelength range. The wavelength-division (de)multiplexers consist of two bi-directional micro-ring resonator arrays for four 16-channel WDM signals. Micro-heaters are placed on the micro-resonators for thermal tuning. According to the experimental results, the excess loss is <3.9 dB in one free spectral range from 1522 nm to 1552 nm and <5.6 dB in three free spectral ranges from 1493 nm to 1583 nm. The intermode crosstalks are −23.2 dB to −33.2 dB, and the isolations between adjacent and nonadjacent wavelength channels are about −17.1 dB and −22.3 dB, respectively. The thermal tuning efficiency is ∼2.22 mW/nm over one free spectral range.

**Keywords:** hybrid (de)multiplexer; wavelength-division multiplexing; mode-division multiplexing; micro-ring; thermal tuning

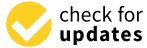



## 1. Introduction

On-chip optical interconnections with ultra-high capacity have been extensively investigated to satisfy the increasing demands for huge data transmissions recently [1,2]. To further improve the link capacity, researchers explore various advanced multiplexing technologies, including wavelength-division multiplexing (WDM) [3–6], mode-division multiplexing (MDM) [7–9], and polarization-division multiplexing (PDM) [10,11]. WDM devices have been developed successfully in the past decades for both long-distance optical fiber communication systems and short-distance optical interconnects. In a WDM system, different wavelengths carry independent signals, thus the cost of lasers and thermal management increases greatly when increasing the wavelength channel number. Meanwhile, multiple channels mean a larger wavelength-band, which makes WDM systems limited by the bandwidth of amplifiers [12]. MDM is provided as a solution to the issues above, which utilizes multimode waveguides to carry independent signals in different modes. Researchers have demonstrated that fiber directional couplers achieve mode conversion off chips [13], however, which are not as low-cost and high-capacity as on-chip MDM. So far, as one of the most crucial components, mode (de)multiplexers have attracted many studies to realize low-loss and low-crosstalk MDM systems [14,15]. PDM also plays a role in multiplexing technologies by making use of dual polarizations to carry signals.

The multiplexing technologies above are based on independent degrees of freedom. Therefore, it is feasible to further enhance link capacity by developing the multi-dimensional hybrid multiplexing technology, which combines two or all of WDM, MDM, and PDM

together [16–21]. In [16], an 8-channel hybrid multiplexer exploiting asymmetric directional couplers (ADCs) was realized to enable simultaneous MDM and PDM. In [17], an 18-channel PDM and WDM hybrid multiplexer has been proposed by utilizing a polarization diversity circuit and single bidirectional arrayed waveguide gratings (AWG). In [19], a 64-channel hybrid MDM and WDM multiplexer consisting of a 4-channel mode demultiplexer and 2 bidirectional AWGs was demonstrated. However, integrating AWGs in WDM systems suffers from phase errors leading to large crosstalks. Later, 32-channel hybrid WDM and MDM multiplexers were proposed and realized by utilizing micro-ring resonator (MRR) arrays in [20,21] for TE and TM modes with a channel spacing of $\Delta\lambda = 3.2$ nm and $\Delta\lambda = 2$ nm, respectively.

In this work, we further extend the capacity by demonstrating a 64-channel hybrid WDM–MDM (de)multiplexer that can work in a broad operational wavelength of three free spectral ranges (FSRs). The 16-channel WDM part is realized with bi-directional MRR arrays. Compared with other filter structures, such as AWGs, MRRs have lower losses and crosstalks and have the potential to realize large-scale integration due to the compact footprint. In addition, we introduce micro-heaters for thermal tuning to align wavelength channels and reduce the impact of fabrication deviation, which suits a smaller channel spacing of 1.6 nm in our design. Here, we utilize MRR arrays bi-directionally to make the device compact and reduce the total power consumption of thermal tuning. Each MRR filter is used to drop the same wavelength channel from two of the mode channels simultaneously. The 4-channel MDM part is implemented with cascaded ADCs, which have a broad wavelength bandwidth, low loss, and compact footprints. For the fabricated 64-channel hybrid WDM–MDM (de)multiplexer, the excess loss is <3.9 dB in one FSR from 1522 nm to 1552 nm and <5.6 dB in three FSRs from 1493 nm to 1583 nm. The intermode crosstalks are $-23.2$dB to $-33.2$ dB, and the isolations between adjacent and nonadjacent wavelength channels are about $-17.1$ dB and $-22.3$ dB, respectively. The thermal tuning efficiency is ~2.22 mW/nm over one FSR.

## 2. Methods

Figure 1a shows the schematic of the 64-channel hybrid WDM–MDM demultiplexer, which can also work as a multiplexer when signals are transmitted in reverse with different lasers. The demultiplexer consists of one 4-channel cascaded ADCs mode demultiplexer and two groups of MRR arrays wavelength demultiplexer. Each group of MRR arrays contains two sets of 16 channels for WDM signals. Micro-ring filters are used bi-directionally, where signals from MDM output-ports $O_1(O_3)$ and $O_2(O_4)$ share the same group of MRR arrays. Figure 1b shows the schematic of the 4-channel cascaded ADCs mode (de)multiplexer. According to the phase-matching condition [22], when effective indices are the same in the access waveguides for the fundamental mode and in the bus waveguide for higher-order mode, the fundamental mode of the access waveguide could be coupled to the desired higher-order mode in the bus waveguide with a judiciously-designed coupling region. To be more specific, the same width (0.41 μm) is chosen for the three access waveguides to make it convenient in our design. The corresponding widths of bus waveguides are chosen as 0.855 μm, 1.3 μm, and 1.75 μm for $TE_1$, $TE_2$, and $TE_3$ modes, respectively, and the gap width between access waveguides and bus waveguides is set to 200 nm. Lengths for coupling regions are calculated as 21 μm, 26 μm, and 31 μm, respectively, according to eigenmode expansion (EME) method [23]. Figure 2 shows the simulated light propagation in the designed ADCs for $TE_1$, $TE_2$, and $TE_3$ modes, respectively. It can be seen that the launched $TE_0$ modes from access waveguides are efficiently coupled to higher-order modes. The fabricated ADCs can be utilized in a broad bandwidth to support 16 channels for WDM with low crosstalks, which are shown in the experimental results section.

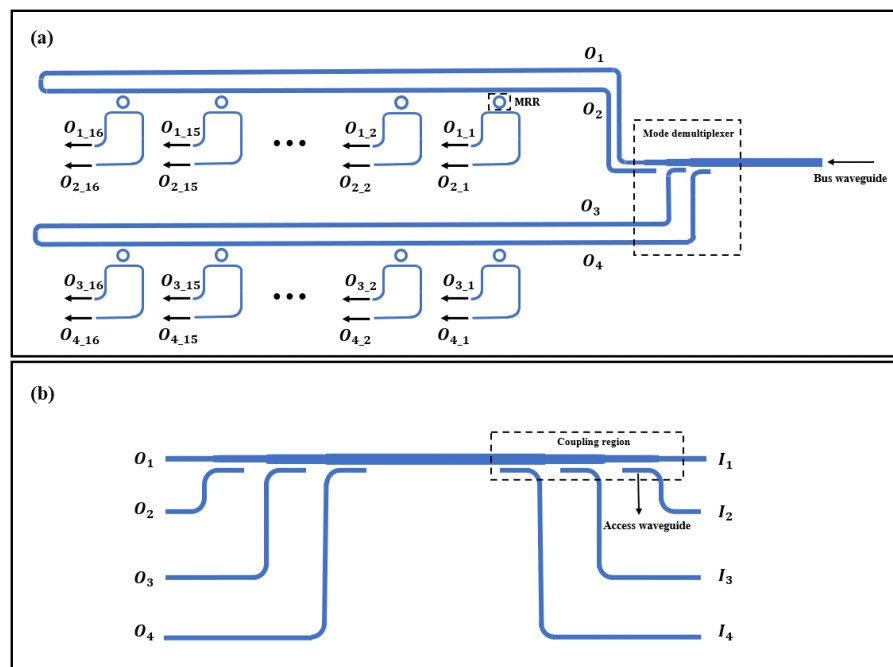

**Figure 1.** Schematic of the (**a**) 64-channel hybrid WDM-MDM demultiplexer, (**b**) 4-channel cascaded ADCs mode (de)multiplexer.

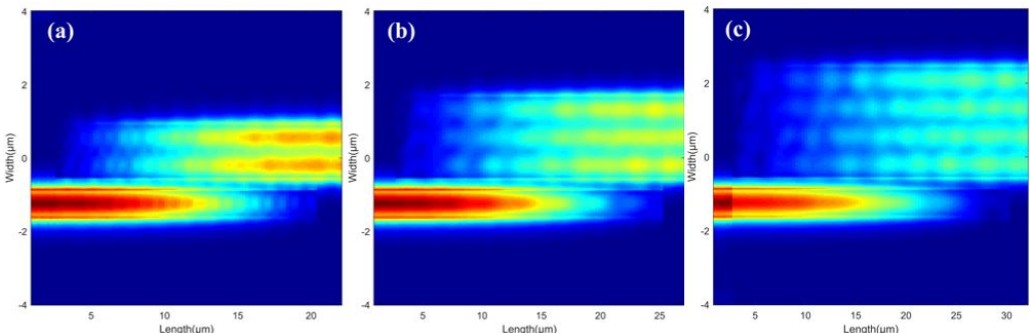

**Figure 2.** Simulated light propagation in the coupling regions of ADCs mode (de)multiplexer when the launched $TE_0$ mode of the access waveguide couples to the (**a**) $TE_1$, (**b**) $TE_2$, and (**c**) $TE_3$ mode in the bus waveguide.

As shown in Figure 1, signals are demultiplexed from the multimode bus waveguide to output-ports $O_i(i = 1, 2, 3, 4)$ of MDM that connect the WDM. To reduce the footprints of the device, two adjacent mode channels share the same MRRs array, and signals are demultiplexed by $j$-th ($j = 1, 2 \ldots, 16$) wavelength channels. For the WDM (de)multiplexer, the wavelength-channel spacing is $\Delta\lambda_{ch} = 1.6$ nm ($\Delta f_{ch} = 200$ GHz @ 1550 nm). Each of the MRR arrays includes 16 MRRs for separating signals. The widths of the MRRs waveguide and bus waveguide are 350 nm and 500 nm, respectively, and the gap between MRRs and the bus waveguide is designed to be 200 nm, which ensures the critical coupling condition. A small micro-ring waveguide bending radius is needed to realize a large FSR to support all multiplexed wavelength channels. In the meanwhile, too small a bending radius will lead to an increase in waveguide loss. In this design, when working in C-band, the radius of MRRs is ~3 μm considering both loss and FSR, and FSR is ~30.1 μm to cover the 16 wavelength channels when $\Delta\lambda_{ch} = 1.6$ nm. For the purpose of the wavelength alignment of all channels, we shift the resonance wavelength of MRRs by thermal tuning, which changes the refractive index and thereby the spectral response due to the large thermo-optic response of silicon [24]. In this design, we utilize titanium nitride micro-heaters and put them on top of the device. The length of each micro-heater is ~90 microns, and the

resistance is as large as ∼1000 ohms to withstand high current and realize a wide tuning range.

## 3. Experimental Results

Figure 3 shows the microscopic image of the fabricated 64-channel hybrid WDM–MDM (de)multiplexer, which combines one 4-channel ADCs MDM and four 16-channel MRRs WDMs. The device was designed and fabricated on a 220 nm SOI platform with silicon dioxide cladding. The demultiplexing process was taken as an example in our experiment, and the multiplexing process can be achieved when exchanging the input and output ports. Figure 4 shows the measured spectral responses of the 4-channel cascaded ADCs MDM. Light was launched from the input ports $I_i$ ($i = 1, 2, 3, 4$), then multiplexed, transmitted, and demultiplexed to the output ports $O_i$. It can be seen from Figure 4 that the excess losses were measured to be 0.5 dB, 1.4 dB, 1.7 dB, and 4.5 dB for $TE_0$, $TE_1$, $TE_2$, and $TE_3$ modes at 1550 nm, respectively. The intermode crosstalks were measured to be $-23.2$ dB to $-33.2$ dB for the four channels at 1550 nm. The realized MDM can work in a broad wavelength range from 1493 nm to 1583 nm to support 16 channels for WDM with low excess losses and low crosstalks. The excess loss of coupling between $TE_0$ and $TE_3$ is larger than that between $TE_0$ and other higher-order modes. The reason might be the deviation between the design and fabrication of waveguides, of which the widths affect coupling efficiency. This issue can be further developed by introducing adiabatic tapers [25] in the coupling regions of mode (de)multiplexers to enhance fabrication tolerance.

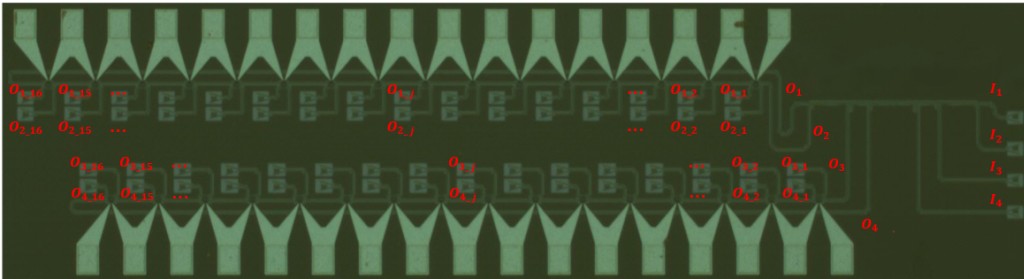

**Figure 3.** Microscopic image of the fabricated 64-channel hybrid WDM–MDM (de)multiplexer.

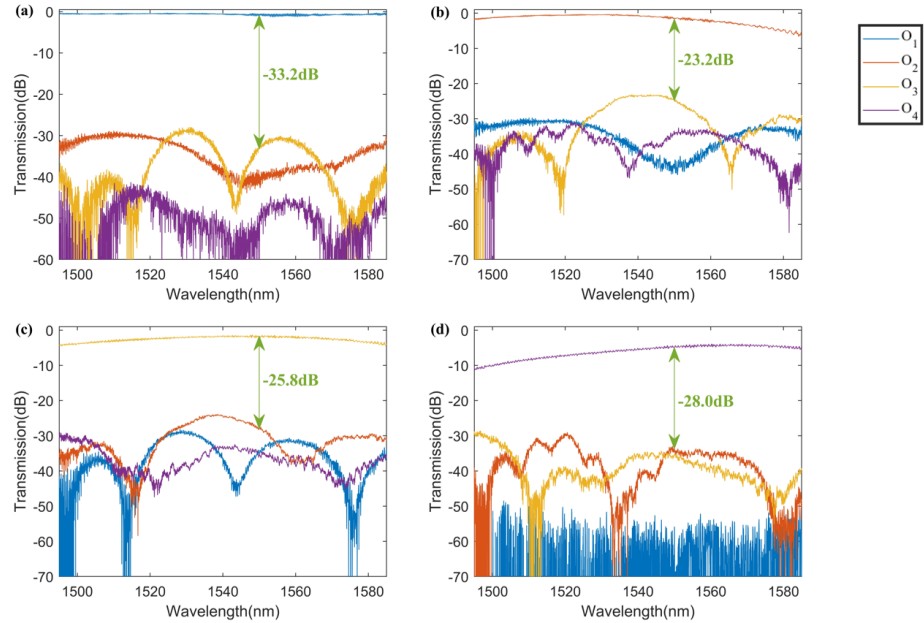

**Figure 4.** Measured spectral responses of the 4-channel cascaded ADCs MDM when light is launched from (**a**) $I_1$, (**b**) $I_2$, (**c**) $I_3$, and (**d**) $I_4$.

Figure 5a shows the microscope image of the fabricated MRR, and the spectral response where light was measured from Input port to Drop port is shown in Figure 5b. It can be seen that the measured excess loss was $\sim -0.2$ dB. The FSR was measured to be $\sim31$ nm, which was large enough to cover 16 channels with $\Delta\lambda_{ch} = 1.6$ nm. According to Figure 5b, the adjacent isolation and non-adjacent isolation of wavelength channels were about $-17.1$ dB and $-22.3$ dB, respectively, for the channel spacing of 1.6 nm.

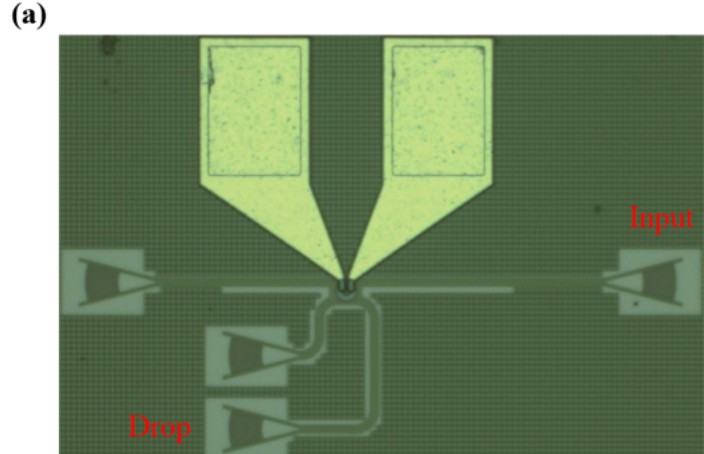

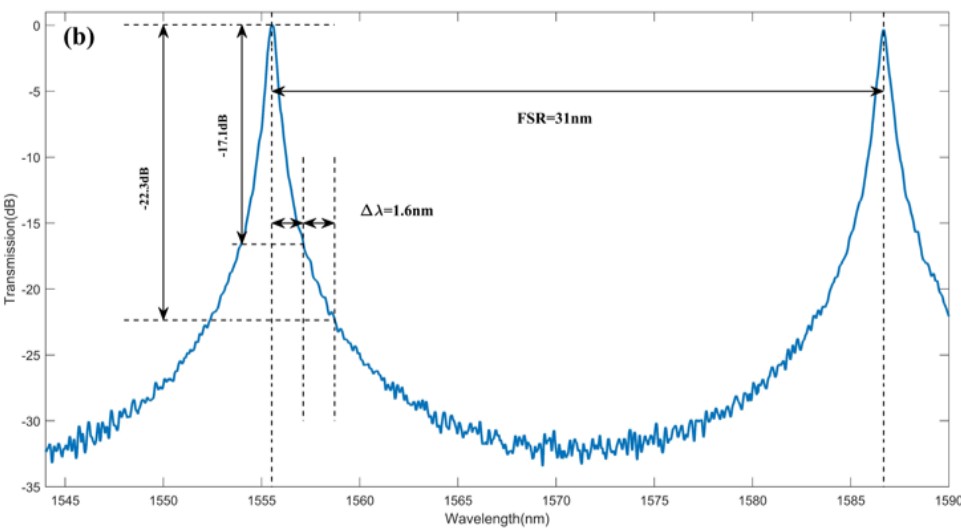

**Figure 5.** (**a**) Microscope image of the fabricated MRR, (**b**) measured spectral response of an MRR.

After signals were demultiplexed from the *i*-th output ports of MDM, all wavelength channels were demultiplexed by the MRRs and dropped to the ports $O_{i\_j}(j = 1, 2\ldots, 16)$. The transmissions of the hybrid MDM–WDM (de)multiplexer were characterized by measuring the spectral responses at each port $O_{i\_j}$ with light injected from $I_i$. We utilized a broad bandwidth light source (1460 nm∼1620 nm) and a polarization controller to obtain TE-polarized incident light. Figure 6 shows the normalized transmission responses by an optical spectrum analyzer. The fabricated device can be used in a broad wavelength range from 1493 nm to 1583 nm. It can be seen that three FSRs of the hybrid MDM–WDM (de)multiplexer were covered, which had an excess loss below 3.9 dB within one FSR from 1522 nm to 1552 nm and below 5.6 dB in three FSRs from 1493 nm to 1583 nm. From Figure 6, it can be seen that when light was injected from input ports $I_i(i = 1, 2, 3, 4)$, the main responses at the drop ports were $O_{i\_j}(j = 1, 2\ldots, 16)$. Some crosstalks might be introduced between two output mode channels that share the same MRR because each

has two drop ports and is used bi-directionally. For example, when signals from MDM output port $O_1$ were not totally dropped by the ports $O_{1\_j}$, the residual undropped signals were transmitted to the port $O_2$, and some signals were reflected at the end of the waveguide of $O_2$ port. Therefore, some signals might be coupled into the same MRRs again and dropped by ports $O_{2\_j}$. In order to suppress the crosstalks mentioned above, taper structures can be designed at the end of ports $O_i$ to make the light gradually dissipate into the cladding to reduce the reflection. The coupling ratio of MRRs can also be optimized to improve the extinction ratio, which results in fewer residual signals. Structures of mode (de)multiplexers can be further researched to obtain lower intermode crosstalks, which can suppress crosstalks between adjacent mode channels.

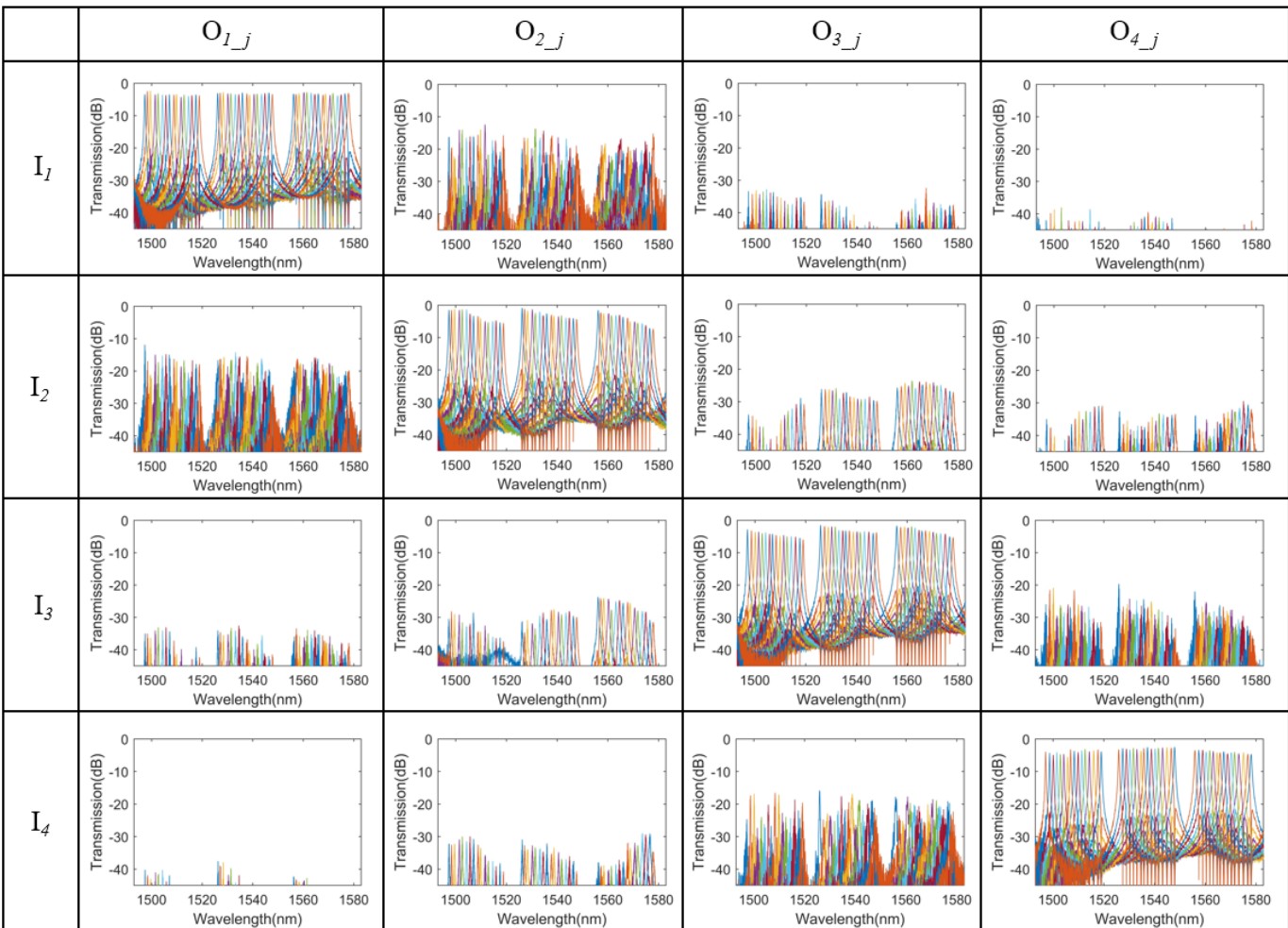

**Figure 6.** Measured spectral responses of the 64-channel hybrid WDM–MDM (de)multiplexer when light is launched from $I_1$, $I_2$, $I_3$, and $I_4$.

In this work, we utilize thermal tuning to achieve more uniform spectral responses, which suits a smaller channel spacing of 1.6 nm in our design. Figure 7a shows the schematic of the micro-heater. A large resistance of the micro-heater as ~1000 ohms is designed to withstand high current and realize a large tuning range. Figure 7b shows spectrum responses at the drop port of one MRR with different applied power. It can be seen that the resonance wavelength of MRR presented a red-shift with increased applied power. One should notice that as the applied power increases, the effective refractive index is changed due to the thermo-optic effect, and the coupling condition might deviate from the critical coupling condition. Therefore, the extinction ratio became smaller under high applied power. Figure 7c shows that resonance wavelength shift exhibited an approximately linear relationship with applied power, where the slope was calculated as 0.45 nm/mW. Figure 7d

shows the transmission of one mode channel where signals were then demultiplexed by WDM channels with 1.6 nm channel spacing in one FSR under thermal tuning. According to the experimental results, the thermal tuning efficiency was ~2.22 mW/nm over one FSR.

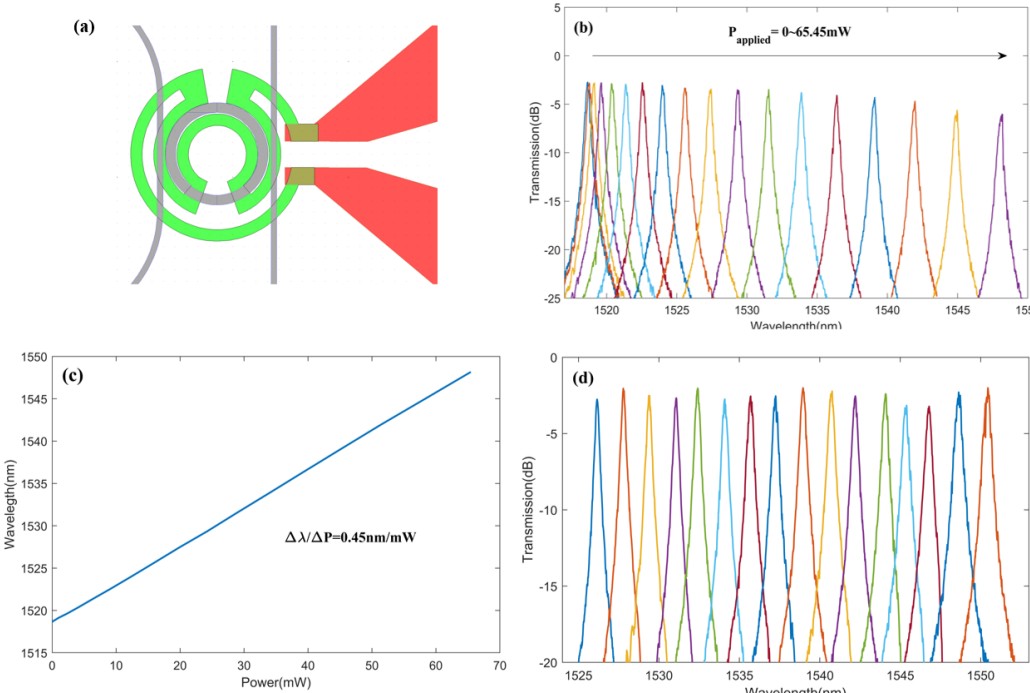

**Figure 7.** (**a**) Schematic of micro-heater (green) and waveguides (grey), (**b**) measured spectral responses of one MRR with different applied power, (**c**) wavelength shift as a linear function of power, (**d**) measured spectral responses of one mode channel where signals are demultiplexed by WDM channels under thermal tuning.

## 4. Discussions and Conclusions

Table 1 gives a comparison of the hybrid MDM–WDM devices. The cascaded ADCs mode (de)multiplexer is not complicated to design, and undesired modes are not excited because of the phase mismatching. According to the experimental results, relatively small intermode crosstalks are realized compared to previous work [8,16]. To enhance fabrication tolerance and reduce the excess losses in a broad wavelength range, adiabatic tapers can be introduced in the coupling region [25]. In this work, an MRR-based WDM structure is investigated to achieve a 64-channel hybrid (de)multiplexer. Compared to [20,21], the capacity of our hybrid (de)multiplexer is further extended by expanding the channel number, and the demonstrated device can operate in a wide wavelength range from 1493 nm to 1583 nm. Compared to other MRR-based WDM structures, the crosstalks of adjacent and non-adjacent wavelength channels reach −17.1 dB and −22.3 dB, which results from a smaller channel spacing of 1.6 nm in this work. Due to the increase in the channel number, thermal tuning is introduced to align wavelength channels that are based on bi-directional MRRs.

In summary, we designed and demonstrated a 64-channel hybrid WDM–MDM (de)mul-tiplexer with a wide operational wavelength from 1493 nm to 1583 nm, and the device was achieved on a 220 nm SOI platform with thermal tuning. Cascaded 4-channel ADCs are introduced in MDM to (de)multiplex and transmit signals in a broad wavelength bandwidth. The intermode crosstalks are −23.2dB to −33.2 dB at 1550 nm. The WDM section utilizes two MRR arrays with a channel spacing of 1.6nm, and FSR is as large as 31 nm to cover 16 channels. The crosstalks between adjacent and non-adjacent wavelength channels are −17.1 dB and −22.3 dB, respectively. For the fabricated hybrid WDM–MDM (de)multiplexer, the excess loss is <3.9 dB in one FSR from 1522 nm to 1552

nm and <5.6 dB in three FSRs from 1493 nm to 1583 nm. The thermal tuning efficiency is ~2.22 mW/nm over one FSR. The performance of the designed hybrid (de)multiplexer can be further enhanced by optimizing the coupling ratio between waveguides and MRRs and by designing a low-crosstalk mode (de)multiplexer. The fabricated 64-channel hybrid (de)multiplexer is useful for boosting link capacity in on-chip interconnections, and the hybrid (de)multiplexer can be extended with more channels and integrated with PDM, which are promising for further improving link capacity in the future.

**Table 1.** Comparison of hybrid WDM–MDM (de)multiplexers.

| WDM Structure | $\Delta\lambda$(nm) | Excess Loss (dB) | Intermode Crosstalk (dB) | Crosstalk of Wavelength Channels (dB) | Channel Number | Thermal Tuning (mW/nm) | Ref. |
|---|---|---|---|---|---|---|---|
| AWG | 3.2 | ~7 | ~−20 | ~−10 | 64 | NA | [18] |
| AWG | 3.2 | 3.5~5.5 | ~−20 | ~−14 | 64 | NA | [19] |
| MRR | 3.2 | 0.5~5 | −16.5 ~−23.5 | −25 ~−35 | 32 | NA | [20] |
| MRR | 2.0 | <4.5 | <−18 | −20~−25 | 32 | NA | [21] |
| MRR | 1.6 | <3.9 in one FSR < 5.6 in three FSR | −23.2~−33.2 | −17.1~−22.3 | 64 | ~2.22 | This work |

**Author Contributions:** Data curation, Y.Y. and H.Y.; Funding acquisition, H.G.; Methodology, H.G. and Y.Y.; Writing—original draft, Y.Y.; Writing—review & editing, Y.Y., H.Y., D.T., X.H., Z.Y., H.G. and Z.L. All authors have read and agreed to the published version of the manuscript.

**Funding:** This research was supported by the National Key Research and Development Program of China (Grant No. 2018YFB2200202) and the National Natural Science Foundation of China (Grant No. 61804148).

**Institutional Review Board Statement:** Not applicable.

**Informed Consent Statement:** Not applicable.

**Data Availability Statement:** The data that support the findings of this study are available from the corresponding author upon reasonable request.

**Conflicts of Interest:** The authors declare no conflict of interest.

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
