# Peer review of "A Silicon-Based On-Chip 64-Channel Hybrid Wavelength- and Mode-Division (de)Multiplexer"

_photonics, doi:10.3390/photonics10020183_

Round 1

Reviewer 1 Report

An on-chip 64-channel hybrid (de)multiplexer for WDM and MDM is designed and demonstrated on a 220 nm SOI plat- form for the demands in large capacity optical interconnections. I think this topic will attract more attentions and this manuscript shows the readers more details. The mode multiplexer is comprised of cascaded asymmetric DC couplers between fundamental and higher-order TE modes, the wavelength-division multiplexers consist of two bi-directional micro-ring resonator arrays for four 16-channel WDM signals.

the authors show us the designed and fabricated device with some experiments, all the results demonstrated the good properties of the device for MDM-WDM interconnections.

Author Response

Point 1: The authors show us the designed and fabricated device with some experiments, all the results demonstrated the good properties of the device for MDM-WDM interconnections.

Response 1: Thank you very much for your approval.

Reviewer 2 Report

The authors demonstrated an on-chip hybrid WDM-MDM (de)multiplexer  with good overall performance. The manuscript is well organized and written. I just would like to give some minor suggestions.

1. In Fig. 1(a), the schematic shows the demultiplexer exactly rather than (de)multiplexer which means multiplexer and demultiplexer. Moreover, O3_1 and O4_1 are marked in wrong symbols.

2. In Fig. 4, O1, O2, O3,O4 are suggested to be marked in different colors to make them clear.

3.Page 4, Line 112. The coupling between TE0 and TE3 is larger than others. Really?

4. An incomplete back-to-back configuration is used in this work as shown in Fig. 3 which lacks the WDM multiplexer. I am curious about the system crosstalk and the influence of transmission length on the crosstalk. Could you please give some comments?

5. Introduction. Simple introduction of the state-of-the-art fiber mode directional couplers are suggested for readers.

Author Response

Point 1:  In Fig. 1(a), the schematic shows the demultiplexer exactly rather than (de)multiplexer which means multiplexer and demultiplexer. Moreover, O3_1 and O4_1 are marked in wrong symbols.

Response 1: Thank you for this valuable question. I am sorry to make Fig.1(a) confusing. Arrows in Fig.1(a) mean that the device works as a demultiplexer, and it can also work as a multiplexer when signals are transmitted in reverse with 64 different lasers. To make our statement clear, we change the sentence ‘Figure 1 (a) shows the schematic of the 64-channel hybrid WDM-MDM (de)multiplexer, which consists of one 4-channel cascaded ADCs mode (de)multiplexer and two groups of MRR arrays wavelength (de)multiplexer.’ to ‘Figure 1(a) shows the schematic of the 64-channel hybrid WDM-MDM demultiplexer, which can also work as a multiplexer when signals are transmitted in reverse with different lasers. The demultiplexer consists of one 4-channel cascaded ADCs mode demultiplexer and two groups of MRR arrays wavelength demultiplexer.’ The name of Fig.1(a) is changed to ’64-channel hybrid WDM-MDM demultiplexer’. Thank you very much for discovering the error about O3_1 and O4_1 symbols, and we correct the symbols in Fig.1(a).

Point 2: In Fig. 4, O1, O2, O3, O4 are suggested to be marked in different colors to make them clear.

Response 2: Thank you very much for this suggestion and we modify Fig.4.

Point 3: Page 4, Line 112. The coupling between TE0 and TE3 is larger than others. Really?

Response 3: Thank you for this valuable question. The excess loss of coupling not the coupling efficiency between TE0 and TE3 is 4.5 dB at 1550 nm, which is larger than that between TE0 and other higher-order modes. We think the reason might be the deviation between the design and fabrication of waveguides. To further reduce the excess losses in a broad wavelength range, adiabatic tapers can be introduced in the coupling regions.

Point 4: An incomplete back-to-back configuration is used in this work as shown in Fig. 3 which lacks the WDM multiplexer. I am curious about the system crosstalk and the influence of transmission length on the crosstalk. Could you please give some comments?

Response 4: Thank you very much for this valuable question. The measurement was taken in the demultiplexing form, where the MRRs arrays parts serve as the WDM demultiplexer. The MRRs arrays can also serve as the WDM multiplexer when Oi_j ports are utilized as input ports with different lasers. To make the statement more rigorous, we add the sentence ‘The demultiplexing process was taken as an example in our experiment, and the multiplexing process can be achieved when exchanging the input and output ports.

There are three main sources of crosstalks in this system. The following statements take the demultiplexing process as an example. First, intermode crosstalks from the MDM part will increase the signals in different Oi ports, which results in system crosstalks. Second, light is not totally dropped in the corresponding MRR, and some crosstalks are introduced when undropped light is coupled into another MRR. Third, we are sorry not to design proper structures at the end of ports O1, O2, O3, and O4 to make the light gradually dissipate into the cladding, so there is reflected light at the end of waveguides due to the sudden change in effective refractive index. This reflected light goes back and is coupled into the same MRRs again. For example, when signals are not totally dropped by the ports O1_j , the residual undropped signals transmit to the port O2, and some signals can be reflected at the end of the waveguide of O2 port. Then, the reflected signals go back and are coupled into the same MRRs. Therefore, this reflection introduces some system crosstalks. The power of signals decreases as the transmission length in the waveguides increases. Thus, the total amount of light coupled into the MRRs reduces, which adversely affects system crosstalks performance.

Point 5: Introduction. Simple introduction of the state-of-the-art fiber mode directional couplers are suggested for readers.

Response 5: Thank you very much for this suggestion, and we add some information about fiber mode directional couplers in the introduction.

Reviewer 3 Report

In the manuscript, the authors design and demonstrate an on-chip 64-channel hybrid (de)multiplexer for wavelength-division multiplexing (WDM) and mode-division multiplexing (MDM) on a 220-nm SOI platform with thermal tuning. The work could be interesting for researchers working for improving link capacity in on-chip interconnections. However, the following comments/questions are suggested to be taken into account before accepting for publication:

1. In lines 77-78, the authors use the eigenmode expansion (EME) method that requires related reference.

2. In lines 157-163, the authors give a comparison of the hybrid MDM-WDM devices shown in Table 1. What are the advantages of the intermode crosstalk in this work compared to previous work? And it is better to add some remarks to explain the reasons.

Author Response

Point 1:  In lines 77-78, the authors use the eigenmode expansion (EME) method that requires related reference.

Response 1: Thank you very much for reminding me of adding the reference, and we add the related reference at the right position.

Point 2: In lines 157-163, the authors give a comparison of the hybrid MDM-WDM devices shown in Table 1. What are the advantages of the intermode crosstalk in this work compared to previous work? And it is better to add some remarks to explain the reasons.

Response 2: Thank you very much for this valuable question. The cascaded ADCs mode (de)multiplexer is not very complicated to be designed and undesired modes will not be excited due to the phase mismatching. Therefore, the intermode crosstalks are low in principle. According to the experimental results, relatively small intermode crosstalks are realized compared to previous work. Intermode crosstalks can be controlled at a low level because significant phase mismatching still exists even with fabrication deviation. To enhance fabrication tolerance and reduce the excess losses in a broad wavelength range, adiabatic tapers can be introduced in the coupling region. Thank you for your suggestion, and we add some remarks in the Discussions and Conclusions section. 

Reviewer 4 Report

In the manuscript by Yuxiang Yin et al., the authors proposed an on-chip 64-channel hybrid (de)multiplexer for wavelength-division multiplexing (WDM) and mode-division multiplexing (MDM). The results show feasible performances for practical application, including the low loss, low crosstalk and small footprint. I recommend the publication on photonics, while I suppose there are still some points in the manuscript need to be addressed.

1, As shown in the microscopic image of the fabricated 64-channel WDM-MDM (de)multiplexer, each channel is coupled with a grating coupler, which is a narrow-band coupling. So how do you measure the device transmission response over a wide band range?

2, According to the measured spectral responses of the 64-channel WDM-MDM (de)multiplexer, there is crosstalk between O1(O3) and O2(O4). Could you explain the reason in more detail? For example, when signals from MDM output port O1 are not totally coupled into drop the ports O1_j , the residual undropped light may transmit to the output port O2, and how the light go back and coupled into the same MRRs again?

3, Following from the second question, how to effectively suppress crosstalk between adjacent mode channels?

Author Response

Point 1: As shown in the microscopic image of the fabricated 64-channel WDM-MDM (de)multiplexer, each channel is coupled with a grating coupler, which is a narrow-band coupling. So how do you measure the device transmission response over a wide band range?

Response 1: Thank you for this valuable question. Although grating couplers do have narrower wavelength bandwidth compared to edge couplers, we can measure relatively small responses where the wavelength deviates from the center wavelength of grating couplers. Then we can obtain normalized transmission responses by subducting responses of two grating couplers. Fig.1 shows the transmission responses of two grating couplers and a very short straight waveguide in this device.

Fig.1 Measured responses of two grating couplers and a very short straight waveguide

Point 2:  According to the measured spectral responses of the 64-channel WDM-MDM (de)multiplexer, there is crosstalk between O1(O3) and O2(O4). Could you explain the reason in more detail? For example, when signals from MDM output port O1 are not totally coupled into drop the ports O1_j , the residual undropped light may transmit to the output port O2, and how the light go back and coupled into the same MRRs again?

Response 2: Thank you very much for this valuable question. I am sorry to make my statement confusing. We did not design proper structures at the end of ports O1, O2, O3, and O4 to make the light gradually dissipate into the cladding, so there is reflected light at the end of waveguides due to the sudden change in effective refractive index. This reflected light goes back and is coupled into the same MRRs again. For example, when signals are not totally dropped by the ports O1_j , the residual undropped signals transmit to the port O2, and some signals can be reflected at the end of the waveguide of O2 port. Then, the reflected signals go back and are coupled into the same MRRs. Waveguides at the end of ports O1, O2, O3, and O4 are marked by red circles in Fig.2. We modify our statement in the passage and mark the new statement with yellow shading.

Fig.2 Waveguides at the end of ports O1, O2, O3, and O4 are marked by red circles

Point 3: Following from the second question, how to effectively suppress crosstalk between adjacent mode channels?

Response 3: Thank you very much for this valuable question. Taper structures can be designed at the end of ports O2, O3, and O4 to make the light gradually dissipate into the cladding to reduce the reflection, then the crosstalk between adjacent mode channels can be suppressed. Furthermore, We can also optimize the coupling ratio of MRRs to improve the extinction ratio, which results in fewer residual signals. Structures of mode (de)multiplexers can be further researched to obtain lower intermode crosstalks, which can also suppress crosstalks between adjacent mode channels.
